# More data speeds up training time in learning halfspaces over sparse vectors

**Amit Daniely**
Department of Mathematics
The Hebrew University
Jerusalem, Israel

**Nati Linial**
School of CS and Eng.
The Hebrew University
Jerusalem, Israel

**Shai Shalev-Shwartz**
School of CS and Eng.
The Hebrew University
Jerusalem, Israel

## Abstract

The increased availability of data in recent years has led several authors to ask whether it is possible to use data as a *computational* resource. That is, if more data is available, beyond the sample complexity limit, is it possible to use the extra examples to speed up the computation time required to perform the learning task?

We give the first positive answer to this question for a *natural supervised learning problem* — we consider agnostic PAC learning of halfspaces over 3-sparse vectors in $\{-1, 1, 0\}^n$. This class is inefficiently learnable using $O\left(n/\epsilon^2\right)$ examples. Our main contribution is a novel, non-cryptographic, methodology for establishing computational-statistical gaps, which allows us to show that, under a widely believed assumption that refuting random 3CNF formulas is hard, it is impossible to efficiently learn this class using only $O\left(n/\epsilon^2\right)$ examples. We further show that under stronger hardness assumptions, even $O\left(n^{1.499}/\epsilon^2\right)$ examples do not suffice. On the other hand, we show a new algorithm that learns this class efficiently using $\tilde{\Omega}\left(n^2/\epsilon^2\right)$ examples. This formally establishes the tradeoff between sample and computational complexity for a natural supervised learning problem.

## 1   Introduction

In the modern digital period, we are facing a rapid growth of available datasets in science and technology. In most computing tasks (e.g. storing and searching in such datasets), large datasets are a burden and require more computation. However, for learning tasks the situation is radically different. A simple observation is that more data can never hinder you from performing a task. If you have more data than you need, just ignore it!

A basic question is how to learn from "big data". The statistical learning literature classically studies questions like "how much data is needed to perform a learning task?" or "how does accuracy improve as the amount of data grows?" etc. In the modern, "data revolution era", it is often the case that the amount of data available far exceeds the information theoretic requirements. We can wonder whether this, seemingly redundant data, can be used for other purposes. An intriguing question in this vein, studied recently by several researchers ([Decatur et al., 1998, Servedio., 2000, Shalev-Shwartz et al., 2012, Berthet and Rigollet, 2013, Chandrasekaran and Jordan, 2013]), is the following

> *Question 1:* Are there any learning tasks in which more data, *beyond the information theoretic barrier*, can *provably* be leveraged to speed up computation time?

The main contributions of this work are:

- Conditioning on the hardness of refuting random 3CNF formulas, we give the first example of a *natural supervised learning problem* for which the answer to Question 1 is positive.

- To prove this, we present a novel technique to establish computational-statistical tradeoffs in supervised learning problems. To the best of our knowledge, this is the first such a result that is not based on cryptographic primitives.

Additional contributions are non trivial efficient algorithms for learning halfspaces over 2-sparse and 3-sparse vectors using $\tilde{O}\left(\frac{n}{\epsilon^2}\right)$ and $\tilde{O}\left(\frac{n^2}{\epsilon^2}\right)$ examples respectively.

The natural learning problem we consider is the task of learning the class of *halfspaces over k-sparse vectors*. Here, the instance space is the *space of k-sparse vectors*,

$$C_{n,k} = \{x \in \{-1,1,0\}^n \mid |\{i \mid x_i \neq 0\}| \leq k\},$$

and the hypothesis class is *halfspaces over k-sparse vectors,* namely

$$\mathcal{H}_{n,k} = \{h_{w,b} : C_{n,k} \to \{\pm 1\} \mid h_{w,b}(x) = \text{sign}(\langle w, x \rangle + b), w \in \mathbb{R}^n, b \in \mathbb{R}\},$$

where $\langle \cdot, \cdot \rangle$ denotes the standard inner product in $\mathbb{R}^n$.

We consider the standard setting of agnostic PAC learning, which models the realistic scenario where the labels are not necessarily fully determined by some hypothesis from $\mathcal{H}_{n,k}$. Note that in the realizable case, i.e. when some hypothesis from $\mathcal{H}_{n,k}$ has zero error, the problem of learning halfspaces is easy even over $\mathbb{R}^n$.

In addition, we allow improper learning (a.k.a. representation independent learning), namely, the learning algorithm is not restricted to output a hypothesis from $\mathcal{H}_{n,k}$, but only should output a hypothesis whose error is not much larger than the error of the best hypothesis in $\mathcal{H}_{n,k}$. This gives the learner a lot of flexibility in choosing an appropriate representation of the problem. This additional freedom to the learner makes it much harder to prove lower bounds in this model. Concretely, it is not clear how to use standard reductions from NP hard problems in order to establish lower bounds for improper learning (moreover, Applebaum et al. [2008] give evidence that such simple reductions do not exist).

The classes $\mathcal{H}_{n,k}$ and similar classes have been studied by several authors (e.g. Long. and Servedio [2013]). They naturally arise in learning scenarios in which the set of all possible features is very large, but each example has only a small number of active features. For example:

- *Predicting an advertisement based on a search query:* Here, the possible features of each instance are all English words, whereas the active features are only the set of words given in the query.

- *Learning Preferences [Hazan et al., 2012]:* Here, we have $n$ players. A *ranking* of the players is a permutation $\sigma : [n] \to [n]$ (think of $\sigma(i)$ as the rank of the $i$'th player). Each ranking induces a *preference* $h_\sigma$ over the ordered pairs, such that $h_\sigma(i,j) = 1$ iff $i$ is ranked higher that $j$. Namely,

$$h_\sigma(i,j) = \begin{cases} 1 & \sigma(i) > \sigma(j) \\ -1 & \sigma(i) < \sigma(j) \end{cases}$$

The objective here is to learn the class, $\mathcal{P}_n$, of all possible preferences. The problem of learning preferences is related to the problem of learning $\mathcal{H}_{n,2}$: if we associate each pair $(i,j)$ with the vector in $C_{n,2}$ whose $i$'th coordinate is 1 and whose $j$'th coordinate is $-1$, it is not hard to see that $\mathcal{P}_n \subset \mathcal{H}_{n,2}$: for every $\sigma$, $h_\sigma = h_{w,0}$ for the vector $w \in \mathbb{R}^n$, given by $w_i = \sigma(i)$. Therefore, every upper bound for $\mathcal{H}_{n,2}$ implies an upper bound for $\mathcal{P}_n$, while every lower bound for $\mathcal{P}_n$ implies a lower bound for $\mathcal{H}_{n,2}$. Since $\text{VC}(\mathcal{P}_n) = n$ and $\text{VC}(\mathcal{H}_{n,2}) = n + 1$, the information theoretic barrier to learn these classes is $\Theta\left(\frac{n}{\epsilon^2}\right)$.

  In Hazan et al. [2012] it was shown that $\mathcal{P}_n$ can be *efficiently* learnt using $O\left(\frac{n \log^3(n)}{\epsilon^2}\right)$ examples. In section 4, we extend this result to $\mathcal{H}_{n,2}$.

We will show a positive answer to Question 1 for the class $\mathcal{H}_{n,3}$. To do so, we show[1] the following:

1. Ignoring computational issues, it is possible to learn the class $\mathcal{H}_{n,3}$ using $O\left(\frac{n}{\epsilon^2}\right)$ examples.

2. It is also possible to *efficiently* learn $\mathcal{H}_{n,3}$ if we are provided with a larger training set (of size $\tilde{\Omega}\left(\frac{n^2}{\epsilon^2}\right)$). This is formalized in Theorem 3.1.

3. It is impossible to *efficiently* learn $\mathcal{H}_{n,3}$, if we are only provided with a training set of size $O\left(\frac{n}{\epsilon^2}\right)$ under Feige's assumption regarding the hardness of refuting random 3CNF formulas [Feige, 2002]. Furthermore, for every $\alpha \in [0, 0.5)$, it is impossible to learn efficiently with a training set of size $O\left(\frac{n^{1+\alpha}}{\epsilon^2}\right)$ under a stronger hardness assumption. This is formalized in Theorem 4.1.

A graphical illustration of our main results is given below:

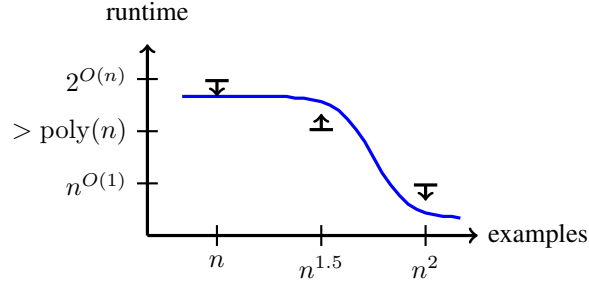

The proof of item 1 above is easy – simply note that $H_{n,3}$ has VC dimension $n + 1$.

Item 2 is proved in section 4, relying on the results of Hazan et al. [2012]. We note, however, that a weaker result, that still suffices for answering Question 1 in the affirmative, can be proven using a naive improper learning algorithm. In particular, we show below how to learn $\mathcal{H}_{n,3}$ efficiently with a sample of $\Omega\left(\frac{n^3}{\epsilon^2}\right)$ examples. The idea is to replace the class $\mathcal{H}_{n,3}$ with the class $\{\pm 1\}^{C_{n,3}}$ containing *all* functions from $C_{n,3}$ to $\{\pm 1\}$. Clearly, this class contains $H_{n,3}$. In addition, we can efficiently find a function $f$ that minimizes the empirical training error over a training set $S$ as follows: For every $x \in C_{n,k}$, if $x$ does not appear at all in the training set we will set $f(x)$ arbitrarily to 1. Otherwise, we will set $f(x)$ to be the majority of the labels in the training set that correspond to $x$. Finally, note that the VC dimension of $\{\pm 1\}^{C_{n,3}}$ is smaller than $n^3$ (since $|C_{n,3}| < n^3$). Hence, standard generalization results (e.g. Vapnik [1995]) implies that a training set size of $\Omega\left(\frac{n^3}{\epsilon^2}\right)$ suffices for learning this class.

Item 3 is shown in section 3 by presenting a novel technique for establishing statistical-computational tradeoffs.

**The class $\mathcal{H}_{n,2}$.** Our main result gives a positive answer to Question 1 for the task of improperly learning $\mathcal{H}_{n,k}$ for $k \geq 3$. A natural question is what happens for $k = 2$ and $k = 1$. Since $\text{VC}(\mathcal{H}_{n,1}) = \text{VC}(\mathcal{H}_{n,2}) = n + 1$, the information theoretic barrier for learning these classes is $\Theta\left(\frac{n}{\epsilon^2}\right)$. In section 4, we prove that $\mathcal{H}_{n,2}$ (and, consequently, $\mathcal{H}_{n,1} \subset \mathcal{H}_{n,2}$) can be learnt using $O\left(\frac{n \log^3(n)}{\epsilon^2}\right)$ examples, indicating that significant computational-statistical tradeoffs start to manifest themselves only for $k \geq 3$.

## 1.1 Previous approaches, difficulties, and our techniques

[Decatur et al., 1998] and [Servedio., 2000] gave positive answers to Question 1 in the realizable PAC learning model. Under cryptographic assumptions, they showed that there exist binary learning problems, in which more data can provably be used to speed up training time. [Shalev-Shwartz et al., 2012] showed a similar result for the agnostic PAC learning model. In all of these papers, the main idea is to construct a hypothesis class based on a one-way function. However, the constructed

---

it is not hard to show that $\mathcal{H}_{n,k}$ can be learnt using a sample of $\Omega\left(\frac{n^k}{\epsilon^2}\right)$ examples by a naive improper learning algorithm, similar to the algorithm we describe in this section for $k = 3$.

classes are of a very synthetic nature, and are of almost no practical interest. This is mainly due to the construction technique which is based on one way functions. In this work, instead of using cryptographic assumptions, we rely on the hardness of refuting random 3CNF formulas. The simplicity and flexibility of 3CNF formulas enable us to derive lower bounds for natural classes such as halfspaces.

Recently, [Berthet and Rigollet, 2013] gave a positive answer to Question 1 in the context of unsupervised learning. Concretely, they studied the problem of sparse PCA, namely, finding a *sparse* vector that maximizes the variance of an unsupervised data. Conditioning on the hardness of the planted clique problem, they gave a positive answer to Question 1 for sparse PCA. Our work, as well as the previous work of Decatur et al. [1998], Servedio. [2000], Shalev-Shwartz et al. [2012], studies Question 1 in the supervised learning setup. We emphasize that unsupervised learning problems are radically different than supervised learning problems in the context of deriving lower bounds. The main reason for the difference is that in supervised learning problems, the learner is allowed to employ improper learning, which gives it a lot of power in choosing an adequate representation of the data. For example, the upper bound we have derived for the class of sparse halfspaces switched from representing hypotheses as halfspaces to representation of hypotheses as tables over $C_{n,3}$, which made the learning problem easy from the computational perspective. The crux of the difficulty in constructing lower bounds is due to this freedom of the learner in choosing a convenient representation. This difficulty does not arise in the problem of sparse PCA detection, since there the learner must output a good sparse vector. Therefore, it is not clear whether the approach given in [Berthet and Rigollet, 2013] can be used to establish computational-statistical gaps in supervised learning problems.

## 2   Background and notation

For hypothesis class $\mathcal{H} \subset \{\pm 1\}^X$ and a set $Y \subset X$, we define the *restriction of $\mathcal{H}$ to $Y$* by $\mathcal{H}|_Y = \{h|_Y \mid h \in \mathcal{H}\}$. We denote by $J = J_n$ the all-ones $n \times n$ matrix. We denote the $j$'th vector in the standard basis of $\mathbb{R}^n$ by $e_j$.

### 2.1   Learning Algorithms

For $h : C_{n,3} \to \{\pm 1\}$ and a distribution $\mathcal{D}$ on $C_{n,3} \times \{\pm 1\}$ we denote the *error of $h$ w.r.t. $\mathcal{D}$* by $\mathrm{Err}_{\mathcal{D}}(h) = \mathrm{Pr}_{(x,y)\sim\mathcal{D}}(h(x) \neq y)$. For $\mathcal{H} \subset \{\pm 1\}^{C_{n,3}}$ we denote the *error of $\mathcal{H}$ w.r.t. $\mathcal{D}$* by $\mathrm{Err}_{\mathcal{D}}(\mathcal{H}) = \min_{h\in\mathcal{H}} \mathrm{Err}_{\mathcal{D}}(h)$. For a sample $S \in (C_{n,3} \times \{\pm 1\})^m$ we denote by $\mathrm{Err}_S(h)$ (resp. $\mathrm{Err}_S(\mathcal{H})$) the error of $h$ (resp. $\mathcal{H}$) w.r.t. the empirical distribution induces by the sample $S$.

A *learning algorithm*, $L$, receives a sample $S \in (C_{n,3} \times \{\pm 1\})^m$ and return a hypothesis $L(S) : C_{n,3} \to \{\pm 1\}$. We say that $L$ *learns $\mathcal{H}_{n,3}$ using $m(n, \epsilon)$ examples* if,[2] for every distribution $\mathcal{D}$ on $C_{n,3} \times \{\pm 1\}$ and a sample $S$ of more than $m(n, \epsilon)$ i.i.d. examples drawn from $\mathcal{D}$,

$$\Pr_S \left(\mathrm{Err}_{\mathcal{D}}(L(S)) > \mathrm{Err}_{\mathcal{D}}(\mathcal{H}_{3,n}) + \epsilon\right) < \frac{1}{10}$$

The algorithm $L$ is *efficient* if it runs in polynomial time in the sample size and returns a hypothesis that can be evaluated in polynomial time.

### 2.2   Refuting random 3SAT formulas

We frequently view a boolean assignment to variables $x_1, \ldots, x_n$ as a vector in $\mathbb{R}^n$. It is convenient, therefore, to assume that boolean variables take values in $\{\pm 1\}$ and to denote negation by "$-$" (instead of the usual "$\neg$"). An $n$-variables 3CNF clause is a boolean formula of the form

$$C(x) = (-1)^{j_1} x_{i_1} \vee (-1)^{j_2} x_{i_2} \vee (-1)^{j_1} x_{i_3}, \quad x \in \{\pm 1\}^n$$

An $n$-variables 3CNF formula is a boolean formula of the form

$$\phi(x) = \wedge_{i=1}^m C_i(x) ,$$

where every $C_i$ is a 3CNF clause. Define the *value*, $\text{Val}(\phi)$, of $\phi$ as the maximal fraction of clauses that can be simultaneously satisfied. If $\text{Val}(\phi) = 1$, we say the $\phi$ is *satisfiable*. By $3\text{CNF}_{n,m}$ we denote the set of 3CNF formulas with $n$ variables and $m$ clauses.

Refuting random 3CNF formulas has been studied extensively (see e.g. a special issue of TCS Dubios et al. [2001]). It is known that for large enough $\Delta$ ($\Delta = 6$ will suffice) a random formula in $3\text{CNF}_{n,\Delta n}$ is not satisfiable with probability $1 - o(1)$. Moreover, for every $0 \le \epsilon < \frac{1}{4}$, and a large enough $\Delta = \Delta(\epsilon)$, the value of a random formula $3\text{CNF}_{n,\Delta n}$ is $\le 1 - \epsilon$ with probability $1 - o(1)$.

The problem of refuting random 3CNF concerns efficient algorithms that provide a proof that a random 3CNF is not satisfiable, or far from being satisfiable. This can be thought of as a game between an adversary and an algorithm. The adversary should produce a 3CNF-formula. It can either produce a satisfiable formula, or, produce a formula uniformly at random. The algorithm should identify whether the produced formula is random or satisfiable.

Formally, let $\Delta : \mathbb{N} \to \mathbb{N}$ and $0 \le \epsilon < \frac{1}{4}$. We say that an efficient algorithm, $A$, $\epsilon$-*refutes random* 3CNF *with ratio* $\Delta$ if its input is $\phi \in 3\text{CNF}_{n,n\Delta(n)}$, its output is either "typical" or "exceptional" and it satisfies:

- *Soundness:* If $\text{Val}(\phi) \ge 1 - \epsilon$, then

$$\Pr_{\text{Rand. coins of } A} \left( A(\phi) = \text{"exceptional"} \right) \ge \frac{3}{4}$$

- *Completeness:* For every $n$,

$$\Pr_{\text{Rand. coins of } A, \ \phi \sim \text{Uni}(3\text{CNF}_{n,n\Delta(n)})} \left( A(\phi) = \text{"typical"} \right) \ge 1 - o(1)$$

By a standard repetition argument, the probability of $\frac{3}{4}$ can be amplified to $1 - 2^{-n}$, while efficiency is preserved. Thus, given such an (amplified) algorithm, if $A(\phi) = \text{"typical"}$, then with confidence of $1 - 2^{-n}$ we know that $\text{Val}(\phi) < 1 - \epsilon$. Since for random $\phi \in 3\text{CNF}_{n,n\Delta(n)}$, $A(\phi) = \text{"typical"}$ with probability $1 - o(1)$, such an algorithm provides, for most 3CNF formulas a proof that their value is less that $1 - \epsilon$.

Note that an algorithm that $\epsilon$-refutes random 3CNF with ratio $\Delta$ also $\epsilon'$-refutes random 3CNF with ratio $\Delta$ for every $0 \le \epsilon' \le \epsilon$. Thus, the task of refuting random 3CNF's gets easier as $\epsilon$ gets smaller. Most of the research concerns the case $\epsilon = 0$. Here, it is not hard to see that the task is getting easier as $\Delta$ grows. The best known algorithm [Feige and Ofek, 2007] 0-refutes random 3CNF with ratio $\Delta(n) = \Omega(\sqrt{n})$. In Feige [2002] it was conjectured that for constant $\Delta$ no efficient algorithm can provide a proof that a random 3CNF is not satisfiable:

**Conjecture 2.1** (R3SAT hardness assumption – [Feige, 2002]). *For every $\epsilon > 0$ and for every large enough integer $\Delta > \Delta_0(\epsilon)$ there exists no efficient algorithm that $\epsilon$-refutes random 3CNF formulas with ratio $\Delta$.*

In fact, for all we know, the following conjecture may be true for every $0 \le \mu \le 0.5$.

**Conjecture 2.2** ($\mu$-R3SAT hardness assumption). *For every $\epsilon > 0$ and for every integer $\Delta > \Delta_0(\epsilon)$ there exists no efficient algorithm that $\epsilon$-refutes random 3CNF with ratio $\Delta \cdot n^\mu$.*

Note that Feige's conjecture is equivalent to the 0-R3SAT hardness assumption.

## 3  Lower bounds for learning $\mathcal{H}_{n,3}$

**Theorem 3.1** (main). *Let $0 \le \mu \le 0.5$. If the $\mu$-R3SAT hardness assumption (conjecture 2.2) is true, then there exists no efficient learning algorithm that learns the class $\mathcal{H}_{n,3}$ using $O\left(\frac{n^{1+\mu}}{\epsilon^2}\right)$ examples.*

In the proof of Theorem 3.1 we rely on the validity of a conjecture, similar to conjecture 2.2 for 3-variables majority formulas. Following an argument from [Feige, 2002] (Theorem 3.2) the validity of the conjecture on which we rely for majority formulas follows the validity of conjecture 2.2.

Define
$$\forall(x_1, x_2, x_3) \in \{\pm 1\}^3, \ \text{MAJ}(x_1, x_2, x_3) := \text{sign}(x_1 + x_2 + x_3)$$
An $n$-variables 3MAJ clause is a boolean formula of the form
$$C(x) = \text{MAJ}((-1)^{j_1} x_{i_1}, (-1)^{j_2} x_{i_2}, (-1)^{j_1} x_{i_3}), \ \ x \in \{\pm 1\}^n$$
An $n$-variables 3MAJ formula is a boolean formula of the form
$$\phi(x) = \wedge_{i=1}^m C_i(x)$$
where the $C_i$'s are 3MAJ clauses. By $3\text{MAJ}_{n,m}$ we denote the set of 3MAJ formulas with $n$ variables and $m$ clauses.

**Theorem 3.2** ([Feige, 2002]). *Let $0 \le \mu \le 0.5$. If the $\mu$-R3SAT hardness assumption is true, then for every $\epsilon > 0$ and for every large enough integer $\Delta > \Delta_0(\epsilon)$ there exists no efficient algorithm with the following properties.*

- *Its input is $\phi \in 3MAJ_{n,\Delta n^{1+\mu}}$, and its output is either* "typical" *or* "exceptional".

- *If* $\text{Val}(\phi) \ge \frac{3}{4} - \epsilon$, *then*

$$\Pr_{\text{Rand. coins of } A} (A(\phi) = \text{"exceptional"}) \ge \frac{3}{4}$$

- *For every $n$,*

$$\Pr_{\text{Rand. coins of } A, \ \phi \sim \text{Uni}(3MAJ_{n,\Delta n^{1+\mu}})} (A(\phi) = \text{"typical"}) \ge 1 - o(1)$$

Next, we prove Theorem 3.1. In fact, we will prove a slightly stronger result. Namely, define the subclass $\mathcal{H}_{n,3}^d \subset \mathcal{H}_{n,3}$, of homogenous halfspaces with binary weights, given by $\mathcal{H}_{n,3}^d = \{h_{w,0} \mid w \in \{\pm 1\}^n\}$. As we show, under the $\mu$-R3SAT hardness assumption, it is impossible to efficiently learn this subclass using only $O\left(\frac{n^{1+\mu}}{\epsilon^2}\right)$ examples.

*Proof idea:* We will reduce the task of refuting random 3MAJ formulas with linear number of clauses to the task of (improperly) learning $\mathcal{H}_{n,3}^d$ with linear number of samples. The first step will be to construct a transformation that associates every 3MAJ clause with two examples in $C_{n,3} \times \{\pm 1\}$, and every assignment with a hypothesis in $\mathcal{H}_{n,3}^d$. As we will show, the hypothesis corresponding to an assignment $\psi$ is correct on the two examples corresponding to a clause $C$ if and only if $\psi$ satisfies $C$. With that interpretation at hand, every 3MAJ formula $\phi$ can be thought of as a distribution $\mathcal{D}_\phi$ on $C_{n,3} \times \{\pm 1\}$, which is the empirical distribution induced by $\psi$'s clauses. It holds furthermore that $\text{Err}_{\mathcal{D}_\phi}(\mathcal{H}_{n,3}^d) = 1 - \text{Val}(\phi)$.

Suppose now that we are given an efficient learning algorithm for $\mathcal{H}_{n,3}^d$, that uses $\kappa \frac{n}{\epsilon^2}$ examples, for some $\kappa > 0$. To construct an efficient algorithm for refuting 3MAJ-formulas, we simply feed the learning algorithm with $\kappa \frac{n}{0.01^2}$ examples drawn from $\mathcal{D}_\phi$ and answer "exceptional" if the error of the hypothesis returned by the algorithm is small. If $\phi$ is (almost) satisfiable, the algorithm is guaranteed to return a hypothesis with a small error. On the other hand, if $\phi$ is far from being satisfiable, $\text{Err}_{\mathcal{D}_\phi}(\mathcal{H}_{n,3}^d)$ is large. If the learning algorithm is proper, then it must return a hypothesis from $\mathcal{H}_{n,3}^d$ and therefore it would necessarily return a hypothesis with a large error. This argument can be used to show that, unless $NP = RP$, learning $\mathcal{H}_{n,3}^d$ with a *proper* efficient algorithm is impossible. However, here we want to rule out *improper* algorithms as well.

The crux of the construction is that if $\phi$ is random, *no algorithm* (even improper and even inefficient) can return a hypothesis with a small error. The reason for that is that since the sample provided to the algorithm consists of only $\kappa \frac{n}{0.01^2}$ samples, the algorithm won't see most of $\psi$'s clauses, and, consequently, the produced hypothesis $h$ will be *independent of them*. Since these clauses are random, $h$ is likely to err on about half of them, so that $\text{Err}_{D_\phi}(h)$ will be close to half!

To summarize we constructed an efficient algorithm with the following properties: if $\phi$ is almost satisfiable, the algorithm will return a hypothesis with a small error, and then we will declare "exceptional", while for random $\phi$, the algorithm will return a hypothesis with a large error, and we will declare "typical".

Our construction crucially relies on the restriction to learning algorithm with a small sample complexity. Indeed, if the learning algorithm obtains more than $n^{1+\mu}$ examples, then it will see most of $\psi$'s clauses, and therefore it might succeed in "learning" even when the source of the formula is random. Therefore, we will declare "exceptional" even when the source is random.

*Proof.* (of theorem 3.1) Assume by way of contradiction that the $\mu$-R3SAT hardness assumption is true and yet there exists an efficient learning algorithm that learns the class $\mathcal{H}_{n,3}$ using $O\left(\frac{n^{1+\mu}}{\epsilon^2}\right)$ examples. Setting $\epsilon = \frac{1}{100}$, we conclude that there exists an efficient algorithm $L$ and a constant $\kappa > 0$ such that given a sample $S$ of more than $\kappa \cdot n^{1+\mu}$ examples drawn from a distribution $\mathcal{D}$ on $C_{n,3} \times \{\pm 1\}$, returns a classifier $L(S) : C_{n,3} \to \{\pm 1\}$ such that

- $L(S)$ can be evaluated efficiently.

- W.p. $\geq \frac{3}{4}$ over the choice of $S$, $\mathrm{Err}_{\mathcal{D}}(L(S)) \leq \mathrm{Err}_{\mathcal{D}}(\mathcal{H}_{n,3}) + \frac{1}{100}$.

Fix $\Delta$ large enough such that $\Delta > 100\kappa$ and the conclusion of Theorem 3.2 holds with $\epsilon = \frac{1}{100}$. We will construct an algorithm, $A$, contradicting Theorem 3.2. On input $\phi \in 3\mathrm{MAJ}_{n,\Delta n^{1+\mu}}$ consisting of the 3MAJ clauses $C_1, \ldots, C_{\Delta n^{1+\mu}}$, the algorithm $A$ proceeds as follows

1. Generate a sample $S$ consisting of $\Delta n^{1+\mu}$ examples as follows. For every clause, $C_k = \mathrm{MAJ}((-1)^{j_1}x_{i_1}, (-1)^{j_2}x_{i_2}, (-1)^{j_3}x_{i_3})$, generate an example $(x_k, y_k) \in C_{n,3} \times \{\pm 1\}$ by choosing $b \in \{\pm 1\}$ at random and letting

$$(x_k, y_k) = b \cdot \left( \sum_{l=1}^{3} (-1)^{j_l} e_{i_l}, 1 \right) \in C_{n,3} \times \{\pm 1\} .$$

For example, if $n = 6$, the clause is $\mathrm{MAJ}(-x_2, x_3, x_6)$ and $b = -1$, we generate the example

$$((0, 1, -1, 0, 0, -1), -1)$$

2. Choose a sample $S_1$ consisting of $\frac{\Delta n^{1+\mu}}{100} \geq \kappa \cdot n^{1+\mu}$ examples by choosing at random (with repetitions) examples from $S$.

3. Let $h = L(S_1)$. If $\mathrm{Err}_S(h) \leq \frac{3}{8}$, return "exceptional". Otherwise, return "typical".

We claim that $A$ contradicts Theorem 3.2. Clearly, $A$ runs in polynomial time. It remains to show that

- If $\mathrm{Val}(\phi) \geq \frac{3}{4} - \frac{1}{100}$, then

$$\Pr_{\text{Rand. coins of } A} (A(\phi) = \text{"exceptional"}) \geq \frac{3}{4}$$

- For every $n$,

$$\Pr_{\text{Rand. coins of } A, \ \phi \sim \mathrm{Uni}(3\mathrm{MAJ}_{n,\Delta n^{1+\mu}})} (A(\phi) = \text{"typical"}) \geq 1 - o(1)$$

Assume first that $\phi \in 3\mathrm{MAJ}_{n,\Delta n^{1+\mu}}$ is chosen at random. Given the sample $S_1$, the sample $S_2 := S \setminus S_1$ is a sample of $|S_2|$ i.i.d. examples which are independent from the sample $S_1$, and hence also from $h = L(S_1)$. Moreover, for every example $(x_k, y_k) \in S_2$, $y_k$ is a Bernoulli random variable with parameter $\frac{1}{2}$ which is independent of $x_k$. To see that, note that an example whose instance is $x_k$ can be generated by exactly two clauses – one corresponds to $y_k = 1$, while the other corresponds to $y_k = -1$ (e.g., the instance $(1, -1, 0, 1)$ can be generated from the clause $\mathrm{MAJ}(x_1, -x_2, x_4)$ and $b = 1$ or the clause $\mathrm{MAJ}(-x_1, x_2, -x_4)$ and $b = -1$). Thus, given the instance $x_k$, the probability that $y_k = 1$ is $\frac{1}{2}$, independent of $x_k$.

It follows that $\mathrm{Err}_{S_2}(h)$ is an average of at least $\left(1 - \frac{1}{100}\right)\Delta n^{1+\mu}$ independent Bernoulli random variable. By Chernoff's bound, with probability $\geq 1 - o(1)$, $\mathrm{Err}_{S_2}(h) > \frac{1}{2} - \frac{1}{100}$. Thus,

$$\mathrm{Err}_S(h) \geq \left(1 - \frac{1}{100}\right)\mathrm{Err}_{S_2}(h) \geq \left(1 - \frac{1}{100}\right) \cdot \left(\frac{1}{2} - \frac{1}{100}\right) > \frac{3}{8}$$

And the algorithm will output "typical".

Assume now that $\mathrm{Val}(\phi) \geq \frac{3}{4} - \frac{1}{100}$ and let $\psi \in \{\pm 1\}^n$ be an assignment that indicates that. Let $\Psi \in \mathcal{H}_{n,3}$ be the hypothesis $\Psi(x) = \mathrm{sign}\left(\langle \psi, x \rangle\right)$. It can be easily checked that $\Psi(x_k) = y_k$ if and only if $\psi$ satisfies $C_k$. Since $\mathrm{Val}(\phi) \geq \frac{3}{4} - \frac{1}{100}$, it follows that

$$\mathrm{Err}_S(\Psi) \leq \frac{1}{4} + \frac{1}{100} \; .$$

Thus,

$$\mathrm{Err}_S(\mathcal{H}_{n,3}) \leq \frac{1}{4} + \frac{1}{100} \; .$$

By the choice of $L$, with probability $\geq 1 - \frac{1}{4} = \frac{3}{4}$,

$$\mathrm{Err}_S(h) \leq \frac{1}{4} + \frac{1}{100} + \frac{1}{100} < \frac{3}{8}$$

and the algorithm will return "exceptional". □

## 4 Upper bounds for learning $\mathcal{H}_{n,2}$ and $\mathcal{H}_{n,3}$

The following theorem derives upper bounds for learning $\mathcal{H}_{n,2}$ and $\mathcal{H}_{n,3}$. Its proof relies on results from Hazan et al. [2012] about learning $\beta$-decomposable matrices, and due to the lack of space is given in the appendix.

**Theorem 4.1.**

- *There exists an efficient algorithm that learns $\mathcal{H}_{n,2}$ using $O\left(\frac{n \log^3(n)}{\epsilon^2}\right)$ examples*

- *There exists an efficient algorithm that learns $\mathcal{H}_{n,3}$ using $O\left(\frac{n^2 \log^3(n)}{\epsilon^2}\right)$ examples*

## 5 Discussion

We formally established a computational-sample complexity tradeoff for the task of (agnostically and improperly) PAC learning of halfspaces over 3-sparse vectors. Our proof of the lower bound relies on a novel, non cryptographic, technique for establishing such tradeoffs. We also derive a new non-trivial upper bound for this task.

**Open questions.** An obvious open question is to close the gap between the lower and upper bounds. We conjecture that $\mathcal{H}_{n,3}$ can be learnt efficiently using a sample of $\tilde{O}\left(\frac{n^{1.5}}{\epsilon^2}\right)$ examples. Also, we believe that our new proof technique can be used for establishing computational-sample complexity tradeoffs for other natural learning problems.

**Acknowledgements:** Amit Daniely is a recipient of the Google Europe Fellowship in Learning Theory, and this research is supported in part by this Google Fellowship. Nati Linial is supported by grants from ISF, BSF and I-Core. Shai Shalev-Shwartz is supported by the Israeli Science Foundation grant number 590-10.

## Footnotes

[1] In fact, similar results hold for every constant $k \geq 3$. Indeed, since $\mathcal{H}_{n,3} \subset \mathcal{H}_{n,k}$ for every $k \geq 3$, it is trivial that item 3 below holds for every $k \geq 3$. The upper bound given in item 1 holds for every $k$. For item 2,

[2] For simplicity, we require the algorithm to succeed with probability of at least $9/10$. This can be easily amplified to probability of at least $1 - \delta$, as in the usual definition of agnostic PAC learning, while increasing the sample complexity by a factor of $\log(1/\delta)$.

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
