[Supplementary Material · appendix.pdf]

# A  Proof of Theorem 4.1

The proof of the theorem relies on results from Hazan et al. [2012] about learning $\beta$-decomposable matrices. Let $W$ be an $n \times m$ matrix. We define the *symmetrization* of $W$ to be the $(n+m) \times (n+m)$ matrix

$$\text{sym}(W) = \begin{bmatrix} 0 & W \\ W^T & 0 \end{bmatrix}$$

We say that $W$ is $\beta$-*decomposable* if there exist positive semi-definite matrices $P, N$ for which

$$\begin{aligned} \text{sym}(W) &= P - N \\ \forall i, P_{ii}, N_{ii} &\leq \beta \end{aligned}$$

Each matrix in $\{\pm 1\}^{n \times m}$ can be naturally interpreted as a hypothesis on $[n] \times [m]$.

We say that a learning algorithm $L$ learns a class $\mathcal{H}_n \subset \{\pm 1\}^{X_n}$ using $m(n, \epsilon, \delta)$ examples if, for every distribution $\mathcal{D}$ on $X_n \times \{\pm 1\}$ and a sample $S$ of more than $m(n, \epsilon, \delta)$ i.i.d. examples drawn from $\mathcal{D}$,

$$\Pr_S \left( \text{Err}_{\mathcal{D}}(L(S)) > \text{Err}_{\mathcal{D}}(\mathcal{H}_n) + \epsilon \right) < \delta$$

Hazan et al. [2012] have proved[3] that

**Theorem A.1.** *Hazan et al. [2012] The hypothesis class of $\beta$-decomposable $n \times m$ matrices with $\pm 1$ entries ban be efficiently learnt using a sample of $O\left( \frac{\beta^2 (n+m) \log(n+m) + \log(1/\delta)}{\epsilon^2} \right)$ examples.*

We start with a generic reduction from a problem of learning a class $\mathcal{G}_n$ over an instance space $X_n \subset \{-1, 1, 0\}^n$ to the problem of learning $\beta(n)$-decomposable matrices. We say that $\mathcal{G}_n$ is *realized* by $m_n \times m_n$ matrices that are $\beta(n)$-decomposable if there exists a mapping $\psi_n : X_n \to [m_n] \times [m_n]$ such that for every $h \in \mathcal{G}_n$ there exists a $\beta(n)$-decomposable $m_n \times m_n$ matrix $W$ for which $\forall x \in X_n, \ h(x) = W_{\psi_n(x)}$. The mapping $\psi_n$ is called a *realization* of $\mathcal{G}_n$. In the case that the mapping $\psi_n$ can be computed in time polynomial in $n$, we say that $\mathcal{G}_n$ is *efficiently realized* and $\psi_n$ is an *efficient realization*. It follows from Theorem A.1 that:

**Corollary A.2.** *If $\mathcal{G}_n$ is efficiently realized by $m_n \times m_n$ matrices that are $\beta(n)$-decomposable then $\mathcal{G}_n$ can be efficiently learnt using a sample of $O\left( \frac{\beta(n)^2 m_n \log(m_n) + \log(1/\delta)}{\epsilon^2} \right)$ examples.*

We now turn to the proof of Theorem 4.1. We start with the first assertion, about learning $\mathcal{H}_{n,2}$. The idea will be to partition the instance space into a disjoint union of subsets and show that the restriction of the hypothesis class to each subset can be efficiently realized by $\beta(n)$-decomposable. Concretely, we decompose $C_{n,2}$ into a disjoint union of five sets

$$C_{n,2} = \cup_{r=-2}^{2} A_n^r$$

where

$$A_n^r = \left\{ x \in C_{n,2} \mid \sum_{i=1}^{n} x_i = r \right\}.$$

In section A.1 we will prove that

**Lemma A.3.** *For every $-2 \leq r \leq 2$, $\mathcal{H}_{n,2}|_{A_n^r}$ can be efficiently realized by $n \times n$ matrices that are $O(\log(n))$-decomposable.*

To glue together the five restrictions, we will rely on the following Lemma, whose proof is given in section A.1.

**Lemma A.4.** *Let $X_1, ..., X_k$ be partition of a domain $X$ and let $H$ be a hypothesis class over $X$. Define $H_i = H|_{X_i}$. Suppose the for every $H_i$ there exist a learning algorithm that learns $H_i$ using $\leq C(d + \log(1/\delta))/\epsilon^2$ examples, for some constant $C \geq 8$. Consider the algorithm $A$ which*

*receives an i.i.d. training set $S$ of $m$ examples from $X \times \{0,1\}$ and applies the learning algorithm for each $H_i$ on the examples in $S$ that belongs to $X_i$. Then, A learns $H$ using at most*

$$\frac{2Ck(d + \log(2k/\delta))}{\epsilon^2}$$

*examples.*

The first part of Theorem 4.1 is therefore follows from Lemma A.3, Lemma A.4 and Corollary A.2.

Having the first part of Theorem 4.1 and Lemma A.4 at hand, it is not hard to prove the second part of Theorem 4.1:

For $1 \leq i \leq n - 2$ and $b \in \{\pm 1\}$ define

$$D_{n,i,b} = \{x \in C_{n,3} \mid x_i = b \text{ and } \forall j < i, \ x_j = 0\}$$

Let $\psi_n : C_{n,3} \to C_{n,2}$ be the mapping that zeros the first non zero coordinate. It is not hard to see that $\mathcal{H}_{n,3}|_{D_{n,i,b}} = \{h \circ \psi_n|_{D_{n,i,b}} \mid h \in \mathcal{H}_{n,2}\}$. Therefore $\mathcal{H}_{n,3}|_{D_{n,i,b}}$ can be identified with $\mathcal{H}_{n,2}$ using the mapping $\psi_n$, and therefore can efficiently learnt using $O\left(\frac{n \log^3(n) + \log(1/\delta)}{\epsilon^2}\right)$ examples (the dependency on $\delta$ does not appear in the statement, but can be easily inferred from the proof). The second part of Theorem 4.1 is therefore follows from the first part of the Theorem and Lemma A.4.

## A.1 Proofs of Lemma A.3 and Lemma A.4

In the proof, we will rely on the following facts. The *tensor product* of two matrices $A \in M_{n \times m}$ and $B \in M_{k \times l}$ is defined as the $(n \cdot k) \times (m \cdot l)$ matrix

$$A \otimes B = \begin{bmatrix} A_{1,1} \cdot B & \cdots & A_{1,m} \cdot B \\ \vdots & \ddots & \vdots \\ A_{n,1} \cdot B & \cdots & A_{m,m} \cdot B \end{bmatrix}$$

**Proposition A.5.** *Let $W$ be a $\beta$-decomposable matrix and let $A$ be a PSD matrix whose diagonal entries are upper bounded by $\alpha$. Then $W \otimes A$ is $(\alpha \cdot \beta)$-decomposable.*

*Proof.* It is not hard to see that for every matrix $W$ and a symmetric matrix $A$,

$$\mathrm{sym}(W) \otimes A = \mathrm{sym}(W \otimes A)$$

Moreover, since the tensor product of two PSD matrices is PSD, if $\mathrm{sym}(W) = P - N$ is a $\beta$-decomposition of $W$, then

$$\mathrm{sym}(W \otimes A) = P \otimes A - N \otimes A$$

is a $(\alpha \cdot \beta)$-decomposition of $W \otimes A$. $\qquad \square$

**Proposition A.6.** *If $W$ is a $\beta$-decomposable matrix, then so is every matrix obtained from $W$ by iteratively deleting rows and columns.*

*Proof.* It is enough to show that deleting one row or column leaves $W$ $\beta$-decomposable. Suppose that $W'$ is obtained from $W \in M_{n \times m}$ by deleting the $i$'th row (the proof for deleting columns is similar). It is not hard to see that $\mathrm{sym}(W')$ is the $i$'th principal minor of $\mathrm{sym}(W)$. Therefore, since principal minors of PSD matrices are PSD matrices as well, if $\mathrm{sym}(W) = P - N$ is $\beta$-decomposition of $W$ then $\mathrm{sym}(W') = [P]_{i,i} - [N]_{i,i}$ is a $\beta$-decomposition of $W'$. $\qquad \square$

**Proposition A.7.** *Hazan et al. [2012] Let $T_n$ be the upper triangular matrix whose all entries in the diagonal and above are $1$, and whose all entries beneath the diagonal are $-1$. Then $T_n$ is $O(\log(n))$-decomposable.*

Lastly, we will also need the following generalization of proposition A.7

**Proposition A.8.** *Let $W$ be an $n \times n$ $\pm 1$ matrix. Assume that there exists a sequence $0 \leq j(1), \ldots, j(n) \leq n$ such that*

$$W_{ij} = \begin{cases} -1 & j \leq j(i) \\ 1 & j > j(i) \end{cases}$$

*Then, $W$ is $O(log(n))$-decomposable.*

*Proof.* Since switching rows of a $\beta$-decomposable matrix leaves a $\beta$-decomposable matrix, we can assume without loss of generality that $j(1) \leq j(2) \leq \ldots \leq j(n)$. Let $J$ be the $n \times n$ all ones matrix. It is not hard to see that $W$ can be obtained from $T_n \otimes J$ by iteratively deleting rows and columns. Combining propositions A.5, A.6 and A.7, we conclude that $W$ is $O(\log(n))$-decomposable, as required. $\square$

We are now ready to prove Lemma A.3

*Proof.* (of Lemma A.3) Denote $\mathcal{A}_n^r = \mathcal{H}_{n,2}|_{A_n^r}$. We split into cases.

*Case 1, r=0:* Note that $A_n^0 = \{e_i - e_j \mid i, j \in [n]\}$. Define $\psi_n : A_n^0 \to [n] \times [n]$ by $\psi_n(e_i - e_j) = (i, j)$. We claim that $\psi_n$ is an efficient realization of $\mathcal{A}_n^0$ by $n \times n$ matrices that are $O(\log(n))$ decomposable. Indeed, let $h = h_{w,b} \in \mathcal{A}_n^0$, and let $W$ be the $n \times n$ matrix $W_{ij} = W_{\psi_n(e_i - e_j)} = h(e_i - e_j)$. It is enough to show that $W$ is $O(\log(n))$-decomposable.

We can rename the coordinates so that

$$w_1 \geq w_2 \geq \ldots \geq w_n \tag{1}$$

From equation (1), it is not hard to see that there exist numbers

$$0 \leq j(1) \leq j(2) \leq \ldots \leq j(n) \leq n$$

for which

$$W_{ij} = \begin{cases} -1 & j \leq j(i) \\ 1 & j > j(i) \end{cases}$$

The conclusion follows from Proposition A.8

*Case 2, r=2 and r=-2:* We confine ourselves to the case $r = 2$. The case $r = -2$ is similar. Note that $A_n^2 = \{e_i + e_j \mid i \neq j \in [n]\}$. Define $\psi_n : A_n^2 \to [n] \times [n]$ by $\psi_n(e_i + e_j) = (i, j)$. We claim that $\psi_n$ is an efficient realization of $\mathcal{A}_n^2$ by $n \times n$ matrices that are $O(\log(n))$ decomposable. Indeed, let $h = h_{w,b} \in \mathcal{A}_n^2$, and let $W$ be the $n \times n$ matrix $W_{ij} = W_{\psi_n(e_i + e_j)} = h(e_i + e_j)$. It is enough to show that $W$ is $O(\log(n))$-decomposable.

We can rename the coordinates so that

$$w_1 \leq w_2 \leq \ldots \leq w_n \tag{2}$$

From equation (2), it is not hard to see that there exist numbers

$$n \geq j(1) \geq j(2) \geq \ldots \geq j(n) \geq 0$$

for which

$$W_{ij} = \begin{cases} -1 & j \leq j(i) \\ 1 & j > j(i) \end{cases}$$

The conclusion follows from Proposition A.8

*Case 3, r=1 and r=-1:* We confine ourselves to the case $r = 1$. The case $r = -1$ is similar. Note that $A_n^1 = \{e_i \mid i \in [n]\}$. Define $\psi_n : A_n^0 \to [n] \times [n]$ by $\psi_n(e_i) = (i, i)$. We claim that $\psi_n$ is an efficient realization of $\mathcal{A}_n^1$ by $n \times n$ matrices that are 3-decomposable (let alone, $\log(n)$-decomposable). Indeed, let $h = h_{w,b} \in \mathcal{A}_n^1$, and let $W$ be the $n \times n$ matrix with $W_{ii} = W_{\psi_n(e_i)} = h(e_i)$ and $-1$ outside the diagonal. It is enough to show that $W$ is 3-decomposable. Since $J$ is 1-decomposable, it is enough to show that $W + J$ is 2-decomposable. However, it is not hard to see that every diagonal matrix $D$ is $(\max_i |D_{ii}|)$-decomposable. $\square$

*Proof.* (of Lemma A.4) Let $S = (x_1, y_1), \ldots, (x_m, y_m)$ be a training set and let $\hat{m}_i$ be the number of examples in $S$ that belong to $X_i$. Given that the values of the random variables $\hat{m}_1, \ldots, \hat{m}_i$ is determined, we have that w.p. of at least $1 - \delta$,

$$\forall i, \quad \mathrm{Err}_{D_i}(h_i) - \mathrm{Err}_{D_i}(h^*) \leq \sqrt{\frac{C(d + \log(k/\delta))}{\hat{m}_i}},$$

where $D_i$ is the induced distribution over $X_i$, $h_i$ is the output of the $i$'th algorithm, and $h^*$ is the optimal hypothesis w.r.t. the original distribution $D$. Define,

$$m_i = \max\{C(d + \log(k/\delta)), \hat{m}_i\}.$$

It follows from the above that we also have, w.p. at least $1 - \delta$, for every $i$,

$$\mathrm{Err}_{D_i}(h_i) - \mathrm{Err}_{D_i}(h^*) \leq \sqrt{\frac{C(d + \log(k/\delta))}{m_i}} =: \epsilon_i.$$

Let $\alpha_i = D\{(x, y) : x \in \mathcal{X}_i\}$, and note that $\sum_i \alpha_i = 1$. Therefore,

$$\mathrm{Err}_D(h_S) - \mathrm{Err}_D(h^*) \leq \sum_i \alpha_i \epsilon_i = \sum_i \sqrt{\alpha_i} \sqrt{\alpha_i \epsilon_i^2}$$

$$\leq \sqrt{\sum_i \alpha_i} \sqrt{\sum_i \alpha_i \epsilon_i^2} = \sqrt{\sum_i \alpha_i \epsilon_i^2}$$

$$= \sqrt{\frac{C(d + \log(k/\delta))}{m}} \sqrt{\sum_i \frac{\alpha_i m}{m_i}}.$$

Next note that if $\alpha_i m < C(d + \log(k/\delta))$ then $\alpha_i m / m_i \leq 1$. Otherwise, using Chernoff's inequality, for every $i$ we have

$$\Pr[m_i < 0.5\alpha_i m] \leq e^{-\alpha_i m/8} \leq e^{-(d + \log(k/\delta))} = e^{-d} \frac{\delta}{k} \leq \frac{\delta}{k}.$$

Therefore, by the union bound,

$$\Pr[\exists i : m_i < 0.5\alpha_i m] \leq \delta.$$

It follows that with probability of at least $1 - \delta$,

$$\sqrt{\sum_i \frac{\alpha_i m}{m_i}} \leq \sqrt{2k}.$$

All in all, we have shown that with probability of at least $1 - 2\delta$ it holds that

$$\mathrm{Err}_D(h_S) - \mathrm{Err}_D(h^*) \leq \sqrt{\frac{2Ck(d + \log(k/\delta))}{m}}.$$

Therefore, the the algorithm learns $\mathcal{H}$ using

$$\leq \frac{2Ck(d + \log(2k/\delta))}{\epsilon^2}$$

examples. □

## Footnotes

[3]The result of Hazan et al. [2012] is more general than what is stated here. Also, Hazan et al. [2012] considered the online scenario. The result for the statistical scenario, as stated here, can be derived by applying standard online-to-batch conversions (see for example Cesa-Bianchi et al. [2001]).