[Reviews · NeurIPS 2013]

Submitted by Assigned_Reviewer_4

On page 2 -- you say that it is impossible to use standard reductions for
proving NP-hardness. I'd say that this issue is a bit more subtle: It
seems that some classes of reduction will not allow basing hardness of
learning on RP \neq NP. (Cite the paper by Applebaum, Barak and Xiao).

You make the point that you are not basing your results on cryptographic ssumptions. However, it is not clear (at least to me)
that the complexity-theoretic assumption you are using is necessarily
weaker than assuming the existence of one-way functions. Possibly, there
is some evidence that the assumption that there are average-case hard
functions in NP is "weaker" than one-way functions exist. There are some
results by Noam Livne indicating such a possibility. You should discuss the relative hardness of assumptions you are using
vs. those that others have used. For example, is it obvious that the
result you have holds under a strong-enough cryptographic assumption such
as factoring is hard? I suspect it should be possible.

For the upper bounds: I think using your techniques, it should be possible
to show that H_{n, k+1} can be learnt using \tilde{O}(n^k/\epsilon^2)
samples? Is that right? Also, do you think it is possible to push the
lower-bound (using a different assumption?) to show that as k gets large
the gap between information-theoretic sample complexity and complexity
increases?

Minor comments:
---------------
1. Line 130: The proof **of** item 1.
2. On line 343, you mean y_k = b right?

--

I think adding a discussion about your complexity assumption (with respect to other such assumptions) will enhance the value of your paper, and will be particularly appealing to a theory audience. In particular, your result shows that "hardness conjectures" in learning appear to be weaker than other conjectures that people have been willing to make in other areas.

I agree that showing a simple concept class for which a separation exists is interesting. In that sense, I really like your result. However, it may help if you clarify explicitly in the writing that you don't necessarily claim your assumption to be weaker than existence of one-way functions (as it may implicitly appear to say so).

Summary: The paper presents a result showing that under a complexity-theoretic
assumption -- no polynomial-time algorithm can learn a specific concept
class (halfspaces) under a class of distributions (low Hamming weight
boolean vectors) -- with sample complexity that is sufficient for
information-theoretic learning. However, if a much larger sample is
provided polynomial time learning is possible.

The paper follows a long line of such works establishing separation
between information-theoretic sample complexity and computational
complexity, starting with Decatur, Goldreich, and Ron.

Submitted by Assigned_Reviewer_5

This paper provides one of the most natural examples of a learning problem for which the problem becomes computationally tractable when given a sufficient amount of data, but is computationally intractable (though still information theoretically tractable) when given a smaller quantity of data. This computational intractability is based on a complexity-theoretic assumption about the hardness of distinguishing satisfiable 3SAT formulas from random ones at a given clause density (more specifically, the 3MAJ variant of the conjecture).

The specific problem considered by the authors is learning halfspaces over 3-sparse vectors. The authors complement their negative results with nearly matching positive results (if one believes a significantly stronger complexity theoretic conjecture-- that hardness persists even for random formulae whose density is n^mu over the satistfiability threshold). Sadly, the algorithmic results are described in the Appendix, and are not discussed. It seems like they are essentially modifications of Hazan et al.'s 2012, though it would be greatly appreciated if the authors included a high-level discussion of the algorithm. Even if no formal proofs of correctness will fit in the body, a description of the algorithm would be helpful.

Overall I like the paper, but suggest that the authors significantly change the presentation. The proof of 3.1 is much more clear than the high-level intuition of the proof given on page 6--and this section can probably be significantly shortened (at no expense of clarity), allowing for some discussion of the algorithmic results of section 4.

typos/suggestions:
Perhaps rephrase 'refuting random 3**' as 'distinguishing high value formulae from random' (???)
p.6 'it would have necessarily return"
"with proper efficient algorithm"
p. 3 "proof item 1"
appendix A 'B(n)-decomposable. -> B(n)-decomposable matrices
Summary: I really like this paper, and believe this result might turn into one of the more classic examples of 'some problems have a different threshold for information theoretic vs computational tractability'.

Submitted by Assigned_Reviewer_7

Summary of paper :

The paper considers the problem learning half-spaces over 3 sparse input in {+1,-1,0}^n. While the information theoretic sample complexity for this problem is of order n/eps^2, assuming the hardness of refuting random 3CNF formulas, it is shown that one cannot efficiently learn using only order n/eps^2 samples. In fact under a stronger version of the same hardness assumption it is shown that it is not possible to efficiently learn using order n^{1+mu}/eps^2 samples for appropriate mu in [0,0.5).
The hardness result is shown to hold for improper learning.

On the other hand it is shown that one can efficiently learn these half spaces with sample of size n^2/eps^2. Thus a true gap in statistical versus computational complexity in learning is shown. It is also shown that the gap between sample complexity of learning information theoretically and sample complexity of efficiently learning is not present while learning 2 sparse vectors.


Comments :


What can be said about extending the positive result of efficiently learning H_{n,3} in n^2/eps^2 to learning H_{n,k}. The result seems to rely mainly on using Hazan et al algorithm to learning H_{n,2} efficiently by using lemma A.4. Is is possible to extend something like lemma A.4 to deal with H_{n,k} ? Whats is the dependence on k in such a case ?




Summary: Overall the paper is well-written and a nice read. While the reduction/proof is fairly straightforward it is quiet insightful. Providing hardness results for improper learning has been notoriously hard as the usual NP hardness type assumptions cant yield results for improper learning problems. This paper provides a new hardness result for improper learning without using the usual cryptographic hardness assumptions and reduction to lattice problems. The simplicity of the hardness assumption and reduction makes me believe that the result could be useful to other learning scenarios too. I believe the work is definitely worth publishing.
Author Feedback

Author rebuttal: Reviewer 4 seems to miss the two main points of the paper:
1) Relation to previous computational-sample tradeoff results: Reviewer 4 criticizes that the result is just another result about computational-sample tradeoff in a long line of such works. All previous results are about highly synthetic hypothesis classes. This fact was emphasized by the authors of these previous papers as a weakness of the approach. Here we consider halfspaces, which is one of the workhorse hypothesis classes nowadays in machine learning.
2) Cryptographic vs. average case complexity: Reviewer 4 complains that it is not clear whether our hardness assumptions are weaker than cryptographic ones. We did not intend to argue that the average case assumption is more plausible than cryptographic assumptions. The reason we emphasized the difference between the two techniques is because it is very difficult to derive results for natural hypothesis classes using the former, while we show how to derive results for the natural hypothesis class of halfspaces using the latter.


Regarding Reviewer 4’s concern about the possibility to use standard reductions from NP hard problems in the context of improper learning: We do not claim that this is impossible. We just say that nobody succeeded to prove a hardness of improper learning based on NP hardness so far. You’re right that our phrasing is confusing
and we will fix it.


To reviews 5 and 7:
Thanks for the kind review.
Thanks also for the suggestion to include the algorithm for the upper bound in the body of the text. Indeed, the upper bound uses Hazan et al method as a black-box, but it requires additional new ideas.